# On Combining DeepSnake and Global Saliency for Detection of Orchard Apples

**Wang Jing [1,2], Wang Leqi [1], Han Yanling [1], Zhang Yun [1,2] and Zhou Ruyan [1,\*]**

1   College of Information Technology, Shanghai Ocean University, Shanghai 201306, China;
    wangjing@shou.edu.cn (W.J.); m180711095@st.shou.edu.cn (W.L.); ylhan@shou.edu.cn (H.Y.);
    y-zhang@shou.edu.cn (Z.Y.)
2   Key Laboratory of Fisheries Information, Ministry of Agriculture, Shanghai 201306, China
\*   Correspondence: ryzhou@shou.edu.cn; Tel.: +15-692-166-808

**Abstract:** For the fast detection and recognition of apple fruit targets, based on the real-time Deep-Snake deep learning instance segmentation model, this paper provided an algorithm basis for the practical application and promotion of apple picking robots. Since the initial detection results have an important impact on the subsequent edge prediction, this paper proposed an automatic detection method for apple fruit targets in natural environments based on saliency detection and traditional color difference methods. Combined with the original image, the histogram backprojection algorithm was used to further optimize the salient image results. A dynamic adaptive overlapping target separation algorithm was proposed to locate the single target fruit and further to determine the initial contour for DeepSnake, in view of the possible overlapping fruit regions in the saliency map. Finally, the target fruit was labeled based on the segmentation results of the examples. In the experiment, 300 training datasets were used to train the DeepSnake model, and the self-built dataset containing 1036 pictures of apples in various situations under natural environment was tested. The detection accuracy of target fruits under non-overlapping shaded fruits, overlapping fruits, shaded branches and leaves, and poor illumination conditions were 99.12%, 94.78%, 90.71%, and 94.46% respectively. The comprehensive detection accuracy was 95.66%, and the average processing time was 0.42 s in 1036 test images, which showed that the proposed algorithm can effectively separate the overlapping fruits through a not-very-large training samples and realize the rapid and accurate detection of apple targets.

**Keywords:** image processing; target detection; saliency detection; DeepSnake instance segmentation; apple images

## 1. Introduction

Apple is the fruit with the largest economic value and one of the few internationally competitive fruits in China [1]. In recent years, the aging of the population has resulted in a large shortage of agricultural labor. The use of smart engineering technology in the agricultural production process can help improve agricultural productivity [2]. Therefore, the research on and application of apple picking robots are of great significance to reduce the labor intensity of agricultural practitioners, free up agricultural labor and improve apple production.

Detection methods for target apples on trees have been studied since the 1970s. Currently, the main methods to solve this problem mainly focus on traditional image processing algorithms, machine learning algorithms, and in recent years, on the fast developing deep learning technology. Traditional algorithms mostly make use of the red and green color differences between apples and background images, combined with different segmentation or edge detection algorithms, to achieve target detection. Si et al. used the normalized chromatic aberration method for fruit segmentation, and the proposed method was used to detect fruit pictures under different light brightness conditions. Finally, the random

circle method was used to label the target fruit. The number of apples was taken as the unit to calculate the evaluation index, and the recognition rate reached 92% [3]. Wang et al., proposed to use an optimized color difference method to segment apples and other fruits, and finally detect fruits through a random circular Hough transform [4]. Zhao et al., proposed using the chromatic aberration model and improved OTSU method for image segmentation. Taking the number of apples as the unit to calculate the evaluation index, the accuracy rate is 84.7% [5]. Duan et al., proposed to segment night images in aCbR fusion space, and proposed an improved three-point circle determination method to identify overlapping fruits based on the segmentation results. Taking pixels as units to evaluate algorithm performance, the accuracy rate is 82.02% [6]. Machine learning is a common research hotspot in the field of artificial intelligence and pattern recognition. In recent years, it has been widely applied in the fields of medicine, aerospace, industry and so on [7–10]. On the basis of ordinary image processing, researchers have combined important algorithms in the field of machine learning and proposed different feasible algorithms. For example, Li et al. fused the segmentation results obtained by the support vector machine (SVM) classifier based on color features and shape features to identify and mark fruit parts. Twenty pictures were tested, and the accuracy rate was 90.8% based on the number of pictures [11]. With the continuous development of research, deep learning technology has become a hot spot in the research of image recognition like [12–16], including apple fruit targets in natural scenes due to its high detection accuracy. Jia et al., fused a K-means clustering algorithm and GA-RBF-LMS neural network to identify and detect 179 apple fruits, and the accuracy rate calculated by the number of apples was 96.95% [17]. Cheng et al., tested 487 test pictures by adopting the improved Lenet convolutional neural network apple target recognition model, and calculated the evaluation index by taking the number of apples as a unit, and the comprehensive recognition rate reached 93.79% [18]. Zhao et al., proposed apple location method based on the You Only Look Once (YOLOv3) deep convolutional neural network. The accuracy rate calculated by the number of apples is 97% [19]. Anna et al., proposed the YOLOv3 and the YOLOv5 algorithms, where the YOLOv3 algorithm, supplemented by the described pre- and post- processing procedures, quite precisely detects both red and green apples. The YOLOv5 algorithm can detect apples quite precisely without any additional techniques, and the accuracy rate calculated by the number of apples was 97.20% [20]. Jia et al., proposed an optimized Mask R-CNN method for the separation of overlapping fruits. The method is tested by a random test set with 120 images, and the precision rate reached 97.31% [21]. According to the experimental results, the target apple detection accuracy based on ordinary image processing algorithm is low, while the target fruit detection algorithm based on deep learning has a higher recognition accuracy. However, the training images required for collection and labeling require a large amount of time and human cost input [22]. In order to achieve high recognition accuracy, most of the algorithm models are complex in structure level, time-consuming in processing, and unable to realize fast recognition and location. Moreover, most studies only focus on a certain type of apple pictures, and the test data set is of a single type, so the algorithm is applicable to a small range. In 2020, Peng et al., proposed the DeepSnake algorithm to iteratively deform the initial contour through deep learning so as to obtain more accurate instance segmentation results [23]. Due to the lack of public data sets, the DeepSnake algorithm is real-time and can effectively extract contour features through circular convolution, without the need to use a large number of training data sets, which effectively reduces the time cost. Therefore, the DeepSnake algorithm is introduced in this paper to detect the target fruit. Since the results of the DeepSnake algorithm are affected by the initial detection box, this paper integrates the improved saliency detection algorithm to identify the fruit part. At the same time, a dynamic adaptive target detection algorithm was proposed to separate the overlapping fruits and obtain the initial labeling results. Based on the above detection results of target fruits, the DeepSnake instance segmentation algorithm was used to optimize the detection results. It can be seen from the experimental results that the proposed algorithm can effectively separate the overlapping fruits with a small

number of training samples, and is less affected by illumination and foliage occlusion, and can detect the target fruits quickly and accurately with good robustness.

## 2. Materials and Methods

The main function of the DeepSnake algorithm is based on deep learning to give the offset that needs to be adjusted for the points on the input initial contour, and iteratively deform the contour to obtain more accurate instance segmentation results [23]. Since the accuracy of the DeepSnake algorithm is affected by the accuracy of the initial detection box, it is very necessary to improve the accuracy of the initial target fruit detection algorithm. The thesis adopts a target fruit detection algorithm with improved saliency calculation and proposes a dynamic adaptive overlapping fruit separation algorithm to obtain the initial detection result for the phenomenon of overlapping fruits. The DeepSnake algorithm is introduced to further optimize the initial detection results. The overall process of the processing method in this article is shown in Figure 1.

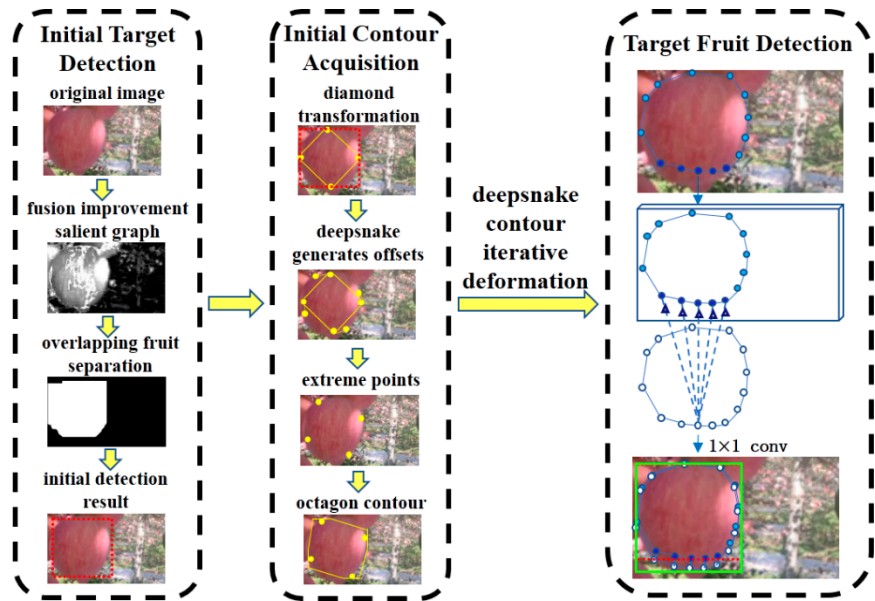

**Figure 1.** The overall flow chart of the proposed method (in the diagram of the DeepSnake processing process, solid round points are input features, triangular points are convolution kernel, and hollow round points are output. The dotted line detection box is the initial detection box, and the solid line detection box is the final detection result obtained after the segmentation of DeepSnake instances).

### 2.1. DeepSnake Network Structure

The DeepSnake network is composed of three parts: the backbone network, the fusion module, and the prediction head. The network structure is shown in Figure 2. The backbone network consists of eight CirConv-Bn-ReLU layers, where CirConv is cyclic convolution, and the batch normalization (Bn) layer refers to the use of a gradient descent algorithm to train the deep neural network, the data of each mini-bacth in the network normalized processing. The rectified linear unit (ReLU) is a non-linear activation unit, and the residual jump connection method is used between all layers. The function of the fusion module is to fuse the multi-scale feature information of all points on the contour. It merges the features of all layers in the backbone network, and then passes them to the $1 \times 1$ convolutional layer, and then performs a maximum pooling operation, so that the fused features are merged with the features of each vertex. The prediction head uses three $1 \times 1$ convolutional layers for the pole features, and then maps them to the offset of each vertex and outputs the offset. Finally, the pole position is moved by the offset, so as to realize the contour deformation.

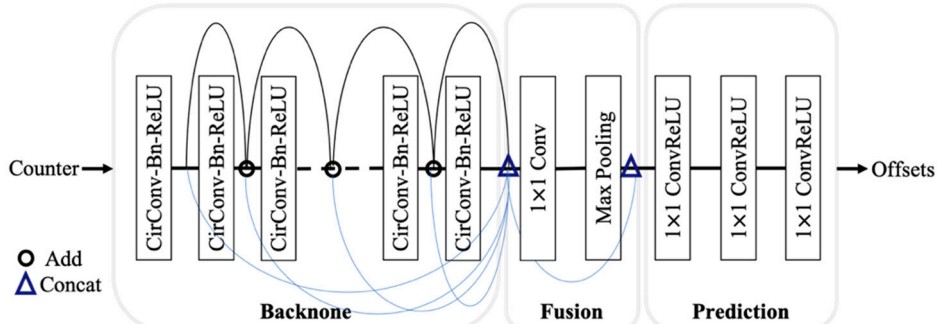

**Figure 2.** DeepSnake network architecture.

### 2.2. Apple Target Detection

Most active contour models require a relatively accurate initial contour. The Deep-Snake algorithm generates four offsets based on the input initial detection box to generate poles $\{x_i^{ex}|i = 1,2,3,4\}$. Based on the poles and the initial detection box, an octagon is generated as the initial contour. The DeepSnake algorithm first samples $N$ points of the initial contour, starting with the top pole $x_1^{ex}$ as the first point and uniformly sampling along the edge. The artificially labeled (Ground truth, GT) contours are also sampled in the same way. The features on the contour are usually regarded as one-dimensional discrete signals and can be processed by standard convolution, but it will destroy the topological structure of the contour. Therefore, the researchers regard the features on the contour as a periodic signal $f_N$, and generate each signal through cyclic convolution. The characteristic vector of each point, the specific formula is as follows:

$$(f_N * k)_i = \sum_{j=-r}^{r} (f_N)_{i+j} k_j \tag{1}$$

where $k$ is the kernel function, and * is the standard convolution operation. Similar to standard convolution, a network layer for feature learning can be constructed on the basis of cyclic convolution. Compared with general convolution, cyclic convolution can make better use of the ring structure of the contour. The feature vector of each point is generated by using cyclic convolution and learning through cyclic convolution. After feature learning, the DeepSnake algorithm uses three $1 \times 1$ convolutional layers to process the output features of each vertex and predict the offset for each point This offset will be used to adjust the contour shape to continuously approach the target. The size of the convolution kernel in the algorithm is 9. The loss function for predicting the poles is as follows:

$$L_{ex} = \frac{1}{4} \sum_{i=1}^{4} l_1(\widetilde{x}_i^{ex} - x_i^{ex}) \tag{2}$$

where $x_i^{ex}$ are the leftmost, rightmost, uppermost, and lowermost poles of the ground truth (GT) edge, and $\widetilde{x}_i^{ex}$ are the predicted poles. The loss function of contour iterative deformation is:

$$L_{iter} = \frac{1}{N} \sum_{i=1}^{N} l_1(\widetilde{x}_i - x_i^{gt}) \tag{3}$$

where $x_i^{ex}$ is the point after contour adjustment, and is the edge point of the corresponding point GT. The detection result is obtained by fusing the improved saliency detection algorithm, and the DeepSnake algorithm iteratively deforms the contour to obtain a more accurate edge, and traverses the edge coordinates to obtain the maximum and minimum values, and further obtains the rectangular detection box coordinates to quickly and accurately label the target fruit and provide a target for the robot arm of the apple picking robot to locate the fruit.

## 3. Target Fruit Detection Based on Fusion Improved Saliency Calculation

Since the accuracy of the DeepSnake algorithm is affected by the accuracy of the initial detection box, it is very necessary to improve the accuracy of the initial target fruit detection algorithm. Peng et al., used CenterNet for the initial detection of the target [23]. This detection algorithm can detect the target object more accurately, but it takes a lot of time and cost to label the data to train the model. Considering that there is currently a lack of public apple data sets in the natural environment, and the initial detection box has a greater impact on the accuracy of the DeepSnake algorithm, this paper combines the classic color difference method, the saliency detection based on the global brightness difference and the histogram back projection algorithm to obtain the saliency map, and further proposes a dynamic adaptive overlapping fruit separation algorithm to separate overlapping fruits and label the target fruits. Considering different color spaces can describe the object from different angles and the lighting conditions of the fruit under natural conditions, in this paper, Lab color space were applied for target feature extraction. First, the RGB color space was changed into Lab color space, performed global L component contrast calculation on the image in the Lab color space and dynamic threshold binarization on the result of saliency detection. After binary processing, the target fruit region was obtained by the maximum connected domain analysis method, and the rectangular region was reduced as a template according to a certain scale. The template was used for RGB histogram back projection. Finally, the result of the histogram back projection was merged with the result of the saliency detection, did binarization processing again. The maximum connected domain is obtained by the maximum connected domain analysis method, and the dynamic number threshold corrosion operation was performed on the largest connected domain, and the corrosion result was obtained. The maximum connected domain analysis was performed again to obtain the maximum connected domain, and finally expanded the maximum connected domain with structural elements such as the number of corrosion operations. The circumscribed rectangle of the largest connected domain was used to achieve the calibration of the fruit. The method can realize fast detection and positioning of the fruit target, provide preconditions for the implementation of the mechanical action of the picking robot, and realize the effective separation of the overlapping fruits, thereby greatly improving the working efficiency of the apple picking robot.

### 3.1. Saliency Map Generation

The Lab model consists of three elements, brightness (L) and two color channels (a and b). The a includes colors from dark green (low brightness value) to gray (middle brightness value) to bright pink (high brightness value); b is from bright blue (low brightness value) to gray (medium brightness value) to yellow (high brightness value). Therefore, this fusion of color will provide with a bright effect. The transformation formulas from RGB to the Lab color space are as follows [24]:

$$\begin{cases} L = 0.607R + 0.174G + 0.201B \\ a = 1.4749(0.2213R - 0.339G + 0.1177B) + 128 \\ b = 0.6245(0.1949R + 0.6057G - 0.8006B) + 128 \end{cases} \qquad (4)$$

Inspired by [25,26] and according to the characteristic of the Lab image, the saliency value of each pixel can be calculated as the sum of the euclidean distances from other pixels in the image. The $L$ channel in the Lab color space was separated, and the image data under the obtained Lab color space is detected by the global $L$ component contrast calculation method to obtain the saliency map $S(I_k)$ under each condition. Thus a saliency map S was generated by the formula as follows:

$$S(I_k) = \sum_{\forall I_k \in I} \| I_k - I_i \| \qquad (5)$$

where $I_i$ is the $L$ channel value of the pixel i, $I_i \in [0, 255]$, $I_k$ is the pixels in the image.

A final saliency map was obtained as follows:

$$S(I_k) = \sum_{i=1}^{n} S(I_k) \tag{6}$$

where $S(I_k)$ to the right of the equal sign is a salient map corresponding to the Lab color space image, i is the number of fused iterations. $n$ is the number of iterations, and the best effect can be achieved when $n$ is 18 through experiments. Image fusion processing can enhance the brightness of the fruit part, which is more distinct between the fruit part and the background.

The RGB color channel is separated, and the color difference is calculated as the pixel value of the result $Se(I_k)$ of this step:

$$Se(I_k) = |I_{kR} - I_{kG}| \tag{7}$$

where $I_{kG}$ is the G channel value of the pixel $I_k$, and $I_{kR}$ is the R channel value of the pixel $I_k$.

The results of color difference method were further fused with the results of saliency detection $(S(I_k)))$ (from Equation (6)) and color difference $(Se(I_k))$ (from Equation (7)) according to the following formula:

$$S(I_k) = S(I_k) + Se(I_k) \tag{8}$$

where $S(I_k)$ to the right of the equal sign is the results of saliency detection of pixsel $I_k$, and $Se(I_k)$ is the result clolor difference of the pixel $I_k$.

We do an open operation on the OTSU [27] result. In this paper, a square with a side length of 27 is used as a structural element for open processing.

### 3.2. Histogram Back Projection

Histogram back projection is used to find the most matching point or region of a particular image [28]. For a A × B test image, there is a template image with a resolution of X × Y. The calculation process of the histogram back projection is as follows:

(1) Cutting a temporary image of (0, 0) to (X, Y) starting from the upper left corner (0, 0) of the input image;
(2) Generating a histogram of the temporary image;
(3) Comparing the histogram of the temporary image with the histogram of the template image, the comparison result of $C(H_1, H_2)$ is calculated from $H'_k(j)$ with Equation (9) and $H'_k(j)$ is calculated using Equation (10):

$$C(H_1, H_2) = \frac{\sum_{i}^{N} H'_1(i) H'_2(i)}{\sqrt{\sum_{i}^{N} H'^2_1(i) H'^2_2(i)}} \tag{9}$$

$$H'_k(i) = H_k(i) - (1/N) \sum_{j}^{N} H_k(j) \tag{10}$$

where $k = 1, 2$, which represents two categories: temporary image and template image. $i = j = 1, 2, 3, \ldots, N$, N is the number of intervals in the histogram, $H_k(i)$ is the ith interval in the kth histogram value. The larger the $C(H_1, H_2)$ value is, the more it matches the template image.

(4) The histogram comparison result c is the pixel value at the result image (0, 0);
(5) Cutting the temporary image of the input image from (0, 1) to (10, Y + 1), comparing the histogram, and recording the result image;
(6) Repeat steps (1) to (5) until the lower right corner of the input image.

In this paper, the circumscribed rectangle was made to the largest connected domain of the binary graph, and the rectangle was reduced to realize the rectangular position inside the apple fruit, and the rectangle was intercepted as a template for the histogram back projection. The rectangle was reduced as follows: x = x + w/2, y = y + h/2, w = w/8, h = h/8, where x, y are the values of the image coordinate (x, y) of the upper left corner of the rectangle, h is the high value of the rectangular box, and w is the value of the rectangular box's width. Target detection result $S_C$ is obtained by histogram backprojection in the original RGB input image. The result of the back projection of the RGB histogram $S_C$ was merged with the salient map($S(I_k)$ (from formula (8)) to obtain the final significance graph $S(I_k)$:

$$S(I_k) = S(I_k) + S_C(I_k) \tag{11}$$

where $S_C$ is the result of back projection.

The addition of the significant detection results to the histogram back projection results, the results of the histogram significance test can fill some of the fruit parts that are not recognized by the significant detection. After finding all the contours, calculating the area of the inner contour, and filling in data when the inner contour area is smaller than the threshold, that is, the pixel value becomes 255, the threshold value in this article is 7000.

*3.3. Overlapping Fruit Separation*

When taking pictures of apples in the natural environment, there will inevitably be overlapping of fruits. In this regard, an adaptive locating method was proposed to separate overlapping fruits. In this paper, a $10 \times 10$ rectangle was used as a structural element to perform multiple etching operations on the binary image, and the maximum connected domain area is calculated after each etching operation. Until the current maximum connected domain area is less than 1/4 times the initial maximum connected domain area, the corrosion operation is stopped. At this time, if there are multiple fruits overlapping, more than one connected domain may appear. Therefore, only the largest connected domain is retained and then expanded by the same number of structural elements for the same number of times to obtain a single target fruit. Based on the binarization results, the target fruit is calibrated by making the circumscribed rectangle the largest connected domain.

## 4. Results and Discussion

Through network resources and on-site collection of 1036 images in the natural environment in Yiyuan County (Zibo City, Shandong Province, China) the test data set was formed. The self-built 1036 images of apples in the natural environment contain a variety of apples. According to the distribution of the fruits, they can be divided into fruits unobstructed by branches and leaves (single unobstructed, multiple apple fruits and adjacent fruits), shaded by branches and leaves, and overlapping fruits. At the same time, the data set contains other external factors that affect fruit recognition, such as patterned labels, plastic bags of apples, poor lighting conditions (large areas of shadows, highlight areas), and fruit with water droplets. The specific structure of the data set is shown in Table 1. This article selects another hundred apple pictures that are different from the test set for image enhancement. By changing the brightness of the picture and flipping the picture horizontally or vertically, the number of training sets is increased to 300, the model is trained, and the network is set. The training batch size is 10, the learning rate is 0.0001, the number of iterations is 150, and the time is 1 h 18 min 55 s. Finally, the self-built 1036 test data sets are used to test the trained model. In this study, the precision rate recall rate (PR) curve, receiver operating characteristic curve (ROC), F-measure (F-measure) [29], overlap (intersection over union, IOU) and recognition accuracy were used to compare the proposed method and other related algorithms were evaluated.

**Table 1.** Class information for the test data set.

|  | Single Fruit | Overlapped Fruit | Connected Fruit | Multiple Fruit | Branch Shade | Total |
|---|---|---|---|---|---|---|
| Poor light conditions | 63 | 225 | 15 | 42 | 52 | 397 |
| Set of plastic bags | 3 | 6 | 3 | 0 | 0 | 12 |
| With water droplets | 36 | 86 | 7 | 14 | 14 | 157 |
| Patterned label | 3 | 6 | 1 | 0 | 0 | 10 |
| The test set | 175 | 556 | 58 | 107 | 140 | 1036 |

*4.1. Experimental Results of Each Stage of the Algorithm*

Aiming at the application scenario where the picking robot automatically detects the apple target on the tree in the natural environment, the DeepSnake algorithm capable of real-time processing is used for instance segmentation. The algorithm effectively learns contour features through cyclic convolution, which further reduces the time cost required for model training. Because the DeepSnake algorithm is sensitive to the initial detection results, it is necessary to improve the accuracy of the initial detection results. Considering that the actual picking time is daytime, the light is sufficient in most cases, and the image brightness feature has a certain influence on the detection of the target. In addition, the time complexity of calculating the brightness contrast difference is less than other saliency detection algorithms [30], which can realize fast processing. On the other hand, the Lab color space can better reflect the color and brightness characteristics of the picture. Therefore, the contrast calculation of the L channel pixel value is calculated as the brightness contrast difference calculation to generate a saliency map. Taking into account the color composition of the picture and the function of the histogram back projection algorithm to detect areas similar to the template image, the saliency detection, color difference method and histogram back projection algorithm are combined to obtain a brighter and complete saliency map of the fruit. Finally, a dynamic adaptive positioning method is used to separate the overlapping fruits, the target fruits are marked with rectangular boxes, and the detection results are further optimized by the DeepSnake algorithm. This paper combines the saliency detection, color difference method and histogram back projection algorithm to obtain the fusion improvement salient graph. Figure 3 shows each stage in the fusion improvement saliency map. It can be seen that from column (a) to column (c), the brightness of the fruit part is increasing and the fruit part is more complete. The improvement of salient graph is not particularly obvious after the histogram backprojection algorithm is integrated. In Figure 3, the optimized salient graph is circled with a rectangular box. Its improvement on the saliency graph is evident in the PR curve. Further from the PR curve in Figure 4, it can be seen that the curves of the prominent image fusion chromatic aberration method and the histogram backprojection algorithm are located at the upper right and the BEP value is largest. Therefore, each step of improving salient graph based on fusion is an optimization of the previous step. Uses dynamic threshold segmentation, opening operation, and hole filling for the saliency map to obtain optimized binary results, and proposes a dynamic adaptive overlapping target separation algorithm to effectively separate overlapping fruits realizes rapid and accurate initial labeling of target fruits, and uses the detection result as the input of the DeepSnake algorithm to obtain the final detection result. The results of each stage of the article algorithm are shown in Figure 5. The dark detection box in the figure is the optimized result of the white initial detection box. On the basis of the initial detection box, the DeepSnake algorithm is introduced to obtain a more accurate fruit edge, and then the circumscribed rectangle of the edge is obtained as the final detection result. It can be seen that the proposed algorithm can effectively separate overlapping fruits and accurately label the target fruits in the original image. For some apple images whose initial detection is not accurate, the introduction of DeepSnake algorithm can significantly optimize the initial detection results.

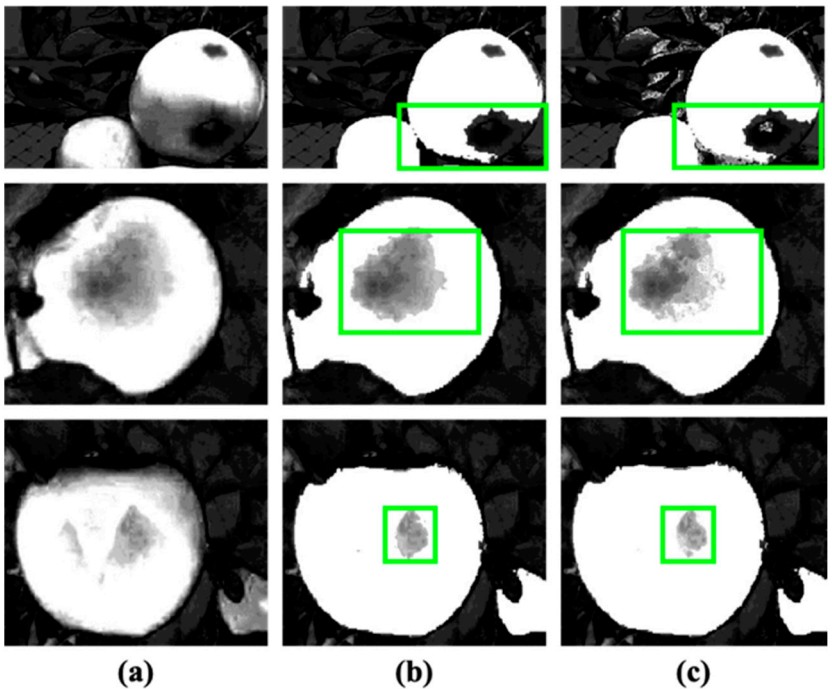

**Figure 3.** Comparison of the results of each stage in the fusion improvement saliency map. (**a**) saliency detection result; (**b**) saliency detection result merged with R-G result; (**c**) saliency detection result merged with R-G result and back projection result.

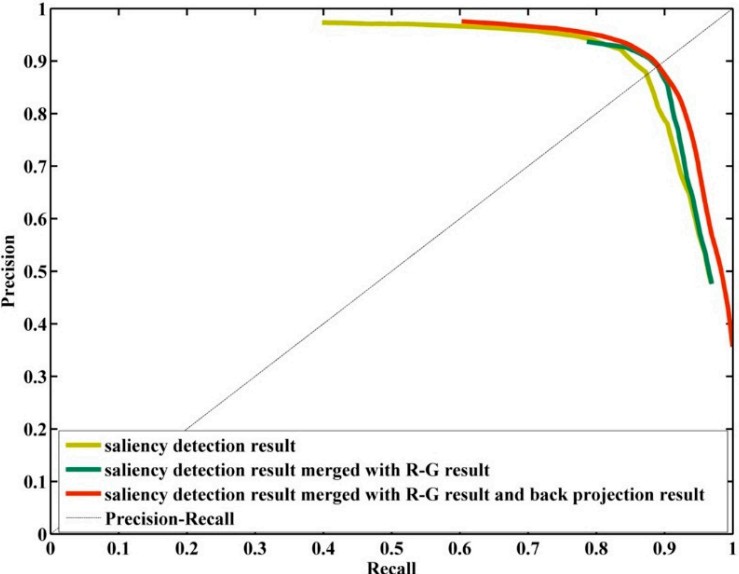

**Figure 4.** PR curve comparison of the results of each stage in the fusion improvement saliency map.

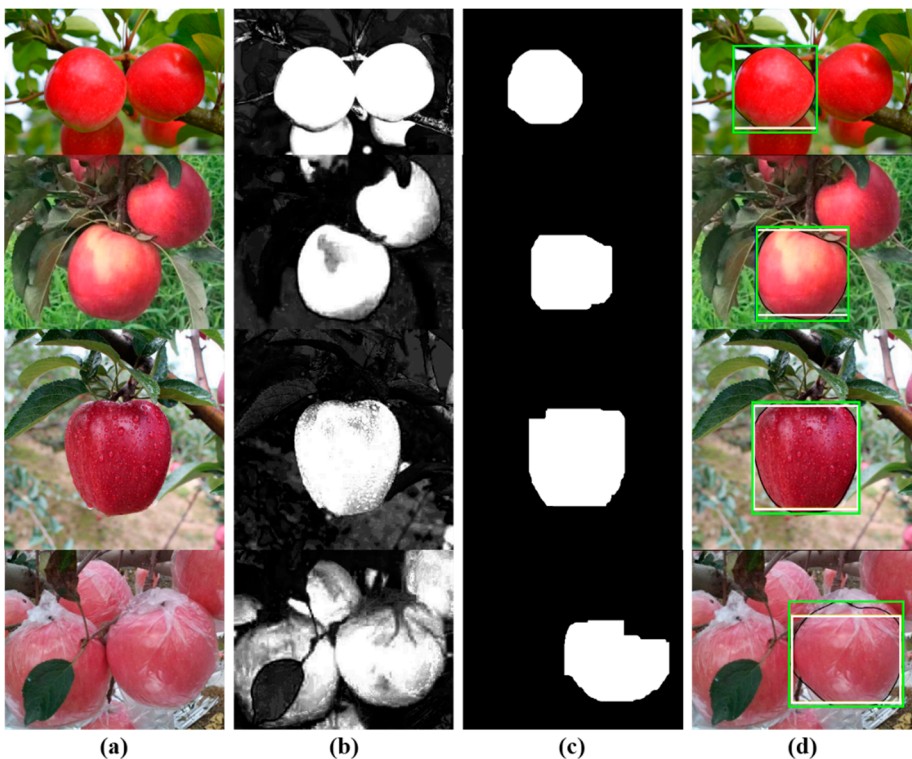

**Figure 5.** Results of each stage of the proposed method. (**a**) Original image; (**b**) Fusion improvement salient graph; (**c**) Overlapped fruit separation results; (**d**) Target fruit detection (the white box in the target fruit labeling is the initial detection result, and the green box is the segmentation optimization result of DeepSnake instance).

*4.2. Comparison and Analysis of Improved Salient Graph and Binary Graph Algorithm*

On 1036 test pictures, the saliency map generated by this method is further combined with the algorithm proposed in [3], the algorithm proposed in [4], the algorithm proposed in [5], and the aCbR fusion space [6] saliency detection algorithm (aCbR saliency detection) and for comparison, some of the processing results are shown in Figure 6. The algorithm proposed in [3] has a large difference in brightness between the fruits and the background, but the individual fruits are not complete. The algorithm proposed in [4] has a relatively complete fruit part, but the background part is brighter, and the brightness difference with the fruit part is small, which is not conducive to subsequent image segmentation. The basic color difference method [5] has a large difference in brightness between the fruit part and the background part, but the edge part of the fruit cannot be completely segmented. The LC saliency detection under the aCbR fusion space [6] is not bright enough for the fruit parts. It can be seen from the results that the method in this paper can accurately separate the apple fruit from the background part. In addition, the fusion-improved saliency detection method proposed in this paper identifies the part of the fruit with higher brightness and more significant. Based on the artificial ground truth (GT) map of manually labeling real apple fruits, the corresponding P-R curve and ROC are generated as shown in Figures 7 and 8. From the perspective of the PR curve, among the PR curves of each algorithm, the curve of the method in this paper is located above the curve of other methods, and is closer to the upper right corner. The curve of the algorithm in the article overlaps and intersects a small part of the curve of the algorithm proposed in [3], but through the balance point (that is, the intersection of the diagonal of the quadrant and the curve), it can be seen that the BEP value of the article algorithm is the largest one, so the result of the algorithm proposed in the article is better than the results of the other four comparison algorithms. In the PR graph, the PR curve of the article algorithm does not intersect the Y axis because the fruit part of the saliency map is too bright, the pixel value is generally close to 255, the

background part is too dark, and the pixel value is generally close to 0, so the curve recall rate is (0, 0.6026) There is no value in the interval. At the same time, it can be seen from the ROC that the curve of the algorithm in this paper is clearly on the upper left of the curve of other algorithms. Therefore, the article algorithm is better than other algorithms. After OTSU adaptive segmentation, mathematical morphology operations and hole filling processing to generate a binary image, and the K-means and GA-RBF-LMS neural network algorithm in the literature [8], the algorithm ((RG)/(R + G)) [3], algorithm (2R-GB) [4], basic color difference method [5], and fusion space [6] saliency detection (cCbR saliency detection + OTSU) segmentation result comparison is shown in Figure 9. Calculated corresponding F-measure values are shown in Table 2. Compared with the other five comparison algorithms, the fruit part of the method in this paper is more complete, with fewer noise points, closer to the GT map, and the F-measure value is the largest. The above qualitative and quantitative processing results show that the performance of the apple initial target detection method in this paper is better than the other five comparison methods.

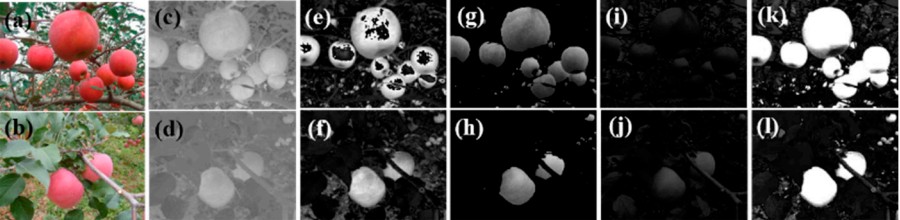

**Figure 6.** Comparison of various apple fruit detection algorithms. (**a,b**) Original image; (**c,d**) algorithm of [3]; (**e,f**) algorithm of [4]; (**g,h**) algorithm of [5]; (**i,j**) aCbR fusion space saliency results; (**k,l**) proposed method.

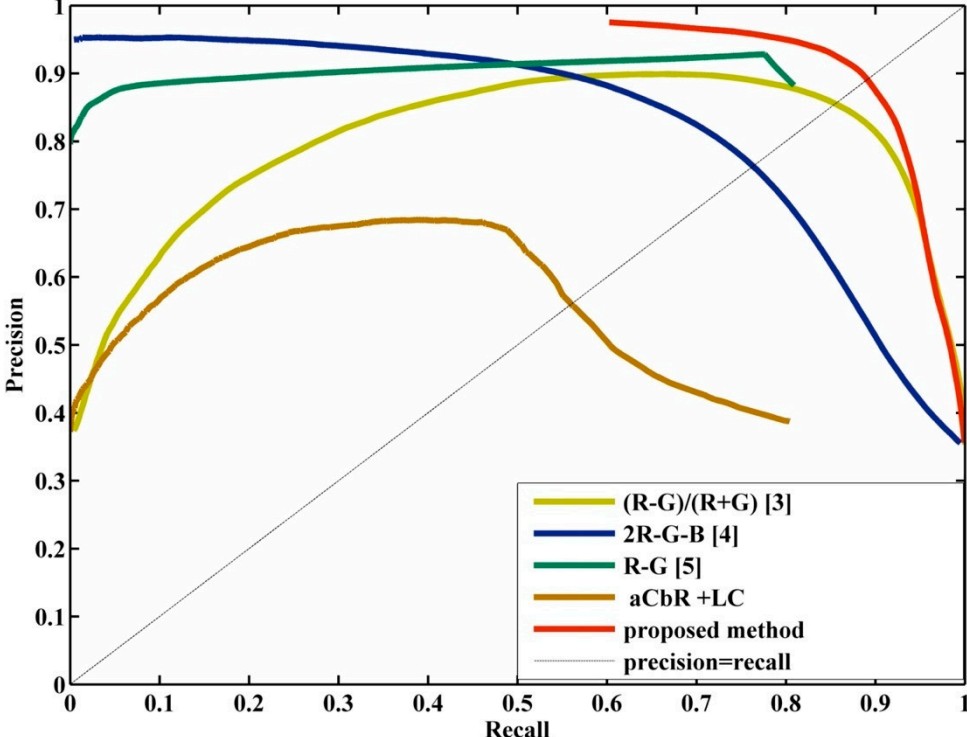

**Figure 7.** P-R curve of different methods.

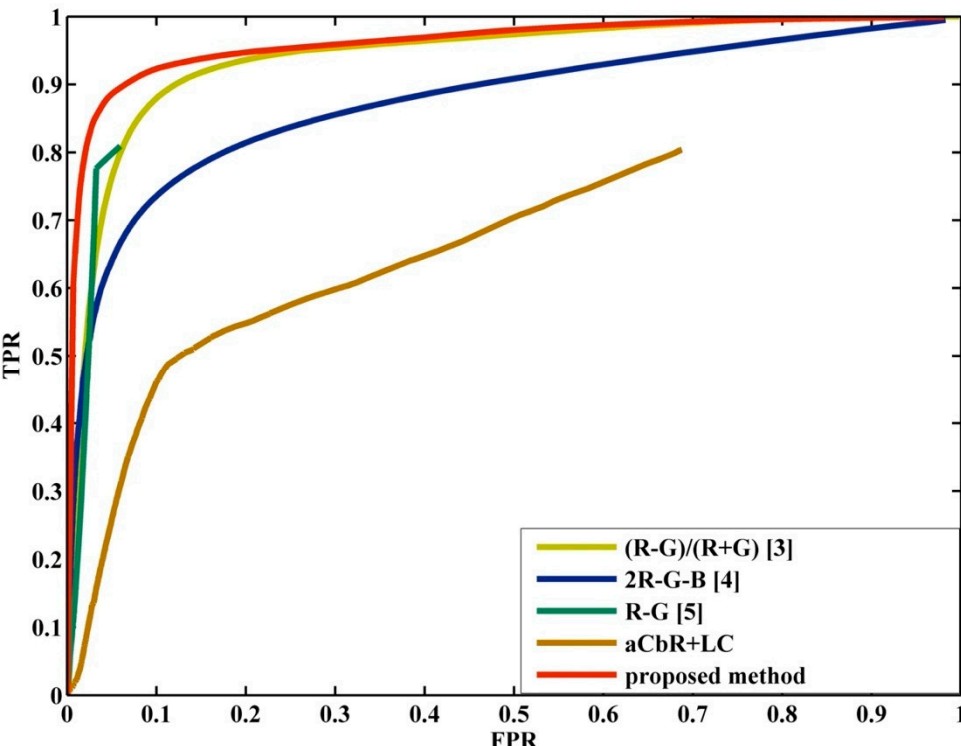

**Figure 8.** ROC of different methods.

**Figure 9.** Comparison of segmentation results by six methods under different conditions. (**a**) individual apple in abundant light; (**b**) overlapped apples in abundant light; (**c**) overlapped apples under low-light condition; (**d**) few apples under low-light condition

**Table 2.** Comparison of F-measure for six methods.

| Method | F-Measure |
|---|---|
| Proposed method | 91.90% |
| (R-G)/(R + G) [3] | 85.98% |
| 2R-G-B [4] | 77.81% |
| R-G [5] | 82.69% |
| cCbR saliency detection + OTSU [6] | 56.27% |
| GA-RBF-LMS [17] | 85.87% |

### 4.3. Comparison and Analysis of Apple Fruit Target Detection Algorithms

The IOU value of the article algorithm is calculated by manually labeling the correct apple position, and the IOU standard deviation is 0.14, and the average value is 0.85. And the calculated accuracy rate of the detection result is 95.66% (accuracy rate = the number of correct test images of the detection result/the number of test sets), which is 5.31% higher than the accuracy of the initial detection result based on the fusion-improved saliency detection. The article algorithm and the detection results of [17], the CenterNet used in [24], the detection results of the DeepSnake algorithm, the detection results of [19], the detection results of [20] and the detection results of [21] are shown in Table 3. It can be seen that the article algorithm combines common image processing algorithms and deep learning algorithms to detect the target fruit, and it can detect the target fruit more accurately with fewer training samples. Although the method based on deep learning can detect the target fruit more accurately, it requires a large number of data sets to train the model. The accuracy of the algorithm depends on the quality of the model, and it takes a lot of time and cost to train the model. Table 4 shows the detection accuracy obtained by testing for overlapping fruits and obscured branches and leaves. From the results, it can be seen that for apple fruits under natural light, the algorithm in this paper can effectively separate overlapping fruits, and can detect the target fruits more accurately in the case of branches and leaves occlusion, poor lighting conditions, etc., and achieve rapid and accurate target labeling. If the number of apples in the picture is more than one, the apple fruit with the largest area in the picture, that is, the apple fruit located closer to the lens, is determined as the target. This also satisfies the actual demand in the picking operation of the apple picking robot.

**Table 3.** Comparison of target fruit detection algorithms in 1036 test data sets.

|  | Precision | The Average IOU | The Standard Deviation of IOU |
|---|---|---|---|
| Proposed method | 95.66% | 0.85 | 0.14 |
| GA-RBF-LMS [17] | 81.37% | 0.70 | 0.29 |
| YOLO v3 [19] | 86.49% | 0.72 | 0.33 |
| YOLO v5 [20] | 87.52% | 0.78 | 0.34 |
| mask R-CNN [21] | 81.39% | 0.72 | 0.28 |
| CenterNet + DeepSnake [24] | 59.65% | 0.56 | 0.46 |

**Table 4.** The accuracy of the test results was measured by the method.

|  | Unobstructed Overlap | Overlapped Fruit | Branches Shade | Poor Light Conditions | Set of Plastic Bags | With Water Droplets | Patterned Label |
|---|---|---|---|---|---|---|---|
| Precision | 99.12% | 94.78% | 90.71% | 94.46% | 91.67% | 98.08% | 80% |

The processing time of the proposed algorithm was evaluated on a self-built 1036 test data set. The resolution of the smallest image contained in the data set is 188 × 186 pixels, and the resolution of the largest picture is 2048 × 1536 pixels. It takes an average of 0.42 s to run the algorithm code for each picture, which can realize fast processing.

### 5. Conclusions

The algorithm presented in this article combines deep learning and starts from multiple angles such as brightness and color information to effectively separate overlapping fruits and achieve rapid and accurate labeling of target fruits. From the detection and labeling results, the accuracy of the proposed algorithm is higher than that of [17] based on K-means and GA-RBF-LMS neural network detection in [19], based on the YOLO deep convolutional neural network algorithm, and in [23], the detection accuracy of CenterNet combined with the DeepSnake Algorithm was 14.29%, 9.17% and 36.01% higher, respec-

tively. From the processing time point of view, the algorithm in this paper can quickly identify and detect the target fruits, which provides the possibility for further application to picking robots. In the future, we will try to replace other neural networks in the DeepSnake algorithm to further improve the detection accuracy and further expand the data set.

**Author Contributions:** Conceptualization, W.J. and W.L.; methodology, W.L.; formal analysis, Z.Y.; investigation, H.Y.; resources, Z.R.; writing-original draft preparation, W.L.; writing-review and editing, W.L. All authors have read and agreed to the published version of the manuscript.

**Funding:** This research was funded by National Natural Science Foundation of China, grant number (61806123 and 41871325); National Key R&D Program of China(2019YFD0900805).

**Institutional Review Board Statement:** Not applicable.

**Informed Consent Statement:** Not applicable.

**Data Availability Statement:** The data presented in this study are available on request from the corresponding author.

**Conflicts of Interest:** The authors declare no conflict of interest.

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
