# Peer review of "On Combining DeepSnake and Global Saliency for Detection of Orchard Apples"

_applsci, doi:10.3390/app11146269_

Round 1

Reviewer 1 Report

Please see the attached report.

Reviewer 2 Report

The article algorithm combines deep learning and starts from multiple angles such as brightness and color information to effectively separate overlapping fruits and achieve rapid and accurate labeling of target fruits.

In general, authors present an algorithm based on deep learning to separate overlapping fruits. Authors should consider the following comments to clarify the main contributions of their paper.  

1.- In the page 1, in the introduction, authors say “Currently, the main methods to solve this problem mainly focus on the traditional image processing algorithm, machine learning algorithm, and the fast developing deep learning technology in recent years.”, in this text, authors should include references [a]-[f] which also use deep learning and machine learning for image processing.

[a] A Fuzzy Logic Model for Hourly Electrical Power Demand Modeling, Electronics, Vol. 10, No. 4, pp. 448, 2021.

[b] Adapting H-Infinity Controller for the Desired Reference Tracking of the Sphere Position in the Maglev Process, Information Sciences, Vol. 569, pp. 669-686, 2021.

[c] Wavelet-Based EEG Processing for Epilepsy Detection Using Fuzzy Entropy and Associative Petri Net, IEEE Access, Vol. 7, pp. 103255-103262, 2019. 

[d] Stability Analysis of the Modified Levenberg-Marquardt Algorithm for the Artificial Neural Network Training, IEEE Transactions on Neural Networks and Learning Systems, 2020. DOI: 10.1109/TNNLS.2020.3015200

[e] On the Estimation and Control of Nonlinear Systems With Parametric Uncertainties and Noisy Outputs, IEEE Access, Vol. 6, pp. 31968-31973, 2018.    

[f] CNN based detectors on planetary environments: a performance evaluation, Frontiers in Neurorobotics, Vol. 14, pp. 85, 2020.  

2.- In the page 6, authors say “A final saliency map was obtained from as follows”, it should be “A final saliency map was obtained as follows”.

3.- In the page 6, 7, in the equations (5)-(8), (11), authors should clarify of these equations represent the structure of the proposed algorithm

4.- In the page 6, in the equations (9), (10), authors should clarify if these equations represent the learning of the proposed algorithm.

5.- In the page 12, in the conclusion, authors should clarify if they explained some future research.

Round 2

Reviewer 1 Report

The paper is acceptable now. One minor comment is that "where" in L211 and other places needs not be indented.